# Resveratrol and Exercise Produce Recovered Ankle and Metatarsus Joint Movements after Penetrating Lesion in Hippocampus in Male Rats

**DOI:** 10.3390/brainsci14100980

**Published:** 2024-09-27

**Authors:** Irene Guadalupe Aguilar-Garcia, Jonatan Alpirez, Rolando Castañeda-Arellano, Judith Marcela Dueñas-Jiménez, Carmen Toro Castillo, Lilia Carolina León-Moreno, Laura Paulina Osuna-Carrasco, Sergio Horacio Dueñas-Jiménez

**Affiliations:** 1Departamento de Neurociencias, Centro Universitario de Ciencias de la Salud, Universidad de Guadalajara, Guadalajara 44340, Mexico; irene.agarcia@academicos.udg.mx (I.G.A.-G.); jonatanalpirez@gmail.com (J.A.); 2Laboratorio de Farmacología, Centro de Investigación Multidisciplinario en Salud, Centro Universitario de Tonalá, Universidad de Guadalajara, Tonalá 45425, Mexico; rolando.castaneda@academicos.udg.mx; 3Departamento de Fisiología, Centro Universitario de Ciencias de la Salud, Universidad de Guadalajara, Guadalajara 44340, Mexico; judith.duenas@academicos.udg.mx; 4Bioingenieria Traslacional, Centro Universitario de Ciencias Exactas e Ingenierías, Universidad de Guadalajara, Guadalajara 44430, Mexico; maria.toro@academicos.udg.mx (C.T.C.); laura.osuna@academicos.udg.mx (L.P.O.-C.); 5Unidad de Evaluación Preclinica, Biotecnología Médica y Farmacéutica, CIATEJ, Guadalajara 44270, Mexico; lileon_pos@ciatej.edu.mx

**Keywords:** locomotion, kinematics, resveratrol, exercise

## Abstract

**Introduction:** This study investigates how traumatic injuries alter joint movements in the ankle and foot. We used a brain injury model in rats, focusing on the hippocampus between the CA1 and dentate gyrus. **Materials and Methods:** We assessed the dissimilarity factor (DF) and vertical displacement (VD) of the ankle and metatarsus joints before and after the hippocampal lesion. We analyzed joint movements in rats after the injury or in rats treated with resveratrol, exercise, or a combination of both. **Results:** Resveratrol facilitated the recovery of DF in both legs, showing improvements in the ankle and metatarsus joints on the third and seventh days post-injury. The hippocampal lesion affected VD in both legs, observed on the third or seventh day after the injury. Both exercise and resveratrol partially recovered VD in the ankle and metatarsus joints on these days. These effects may be linked to increased hippocampal neurogenesis and reduced neuroinflammation. **Conclusions:** The study highlights the benefits of resveratrol and exercise in motor recovery following brain injury, suggesting their potential to enhance the quality of life for patients with neurological disorders affecting motor function and locomotion. These findings also suggest that resveratrol could offer a promising or complementary alternative in managing chronic pain and inflammation associated with orthopedic conditions, thus improving overall patient management.

## 1. Introduction

Traumatic brain injury (TBI) is one of the leading causes of death and severe disability and is also associated with mental health disorders in humans [1]. TBI involves complex human mechanisms, challenging their precise replication in animal models. Therefore, no animal model can mimic human brain injury [2]. However, these models offer the best alternative for investigating the biomechanical, cellular, and molecular mechanisms involved in the neuropathological progression associated with TBI and time-dependent effects [3,4]. The lesion in the hippocampus is associated with motor alterations [5]. We have developed a murine model to investigate the impact of a penetrating lesion in the hippocampus and its consequences on locomotion speed. This lesion not only alters locomotion kinematics, as reported by Lopez Ruiz [6], but also induces changes in the vertical displacement of the ankle and metatarsal joints and the dissimilarity factor [7].

On the other hand, exercise has been considered an indispensable factor in facilitating synaptic plasticity. Exercise promotes neurogenesis, neuronal survival, and regeneration [8,9,10,11]. It also modulates the inflammatory response [12], particularly in two hippocampal regions: the subgranular zone and dentate gyrus. In rats, it has been described to enhance neuronal plasticity [13,14] and inhibit apoptosis in several ischemia models [15]. Other findings from animal studies have contributed to considering exercise as a potential non-pharmacological approach to aid recovery after TBI.

Resveratrol is a flavonoid that benefits health, reduces damage in the hippocampus after ischemia [16], and decreases epileptic episodes [17]. This flavonoid possesses antioxidant and anti-inflammatory effects [18,19]. It causes a reduction in cerebral edema with improvement in cognitive processes and a decrease in motor deficit after TBI [20]. In this study, we implemented a penetrating injury to induce damage in the hippocampus. The effects of exercise and resveratrol in the metatarsus and ankle joint vertical displacement and the dissimilarity factor in rats were analyzed to study the recovery of these joint movements.

## 2. Materials and Methods

### 2.1. Animals and Experimental

A total of 25 adult Wistar rats (180–250 g) were handled under EU Directive 2010/63/EU for animal experiments and Mexican Regulation of Animal Care and Maintenance (NOM-062-ZOO-1999, 2001). Animals were kept on a 12 h light/dark cycle at 22 °C ± 2 °C and constant humidity, with water and food available ad libitum. The groups were divided into five. One group was considered control and only filmed on the third and seventh days without injury and treatment. The second group was lessoned and filmed on the third and seventh day. The third group was injured, administered with resveratrol, and filmed on the third and seventh day. The fourth group was taught and trained in exercise and filmed on the third and seventh days. The fifth group was taught and administered with resveratrol, trained with exercise, filmed on the third and seventh days, and then sacrificed. The same animals were used as the control group. To obtain reliable results, we used a minimal number of animals and minimized their suffering.

### 2.2. Penetrating Injury Model

The animals were anesthetized with 5% isoflurane for induction, then 3%. A cleft was drilled in the left parietal bone to expose the meninges. A 0.5 mm sterile steel cannula was placed at coordinates from Bregma (ML: −2 mm, AP: −5 mm) as per Paxinos and Watson. The cannula penetrated 4 mm, displaced roughly 2 mm, and was removed. After the surgery, the animals were given antibiotics (Enroxil 2 mg/kg) and analgesics (meloxicam 2 mg/kg) for three days.

### 2.3. Resveratrol Treatment

Resveratrol (Santa Cruz #SC-200808, Dallas, TX, USA) was given at 100 mg/kg dissolved in a 50% ethanol and 50% saline solution (0.9% NaCl) via intraperitoneal injection starting the same day of the injury and continuing daily for seventh days post-injury, taken from Sönmez U et al. (2007) [20].

### 2.4. Exercise Training

Exercise training was conducted on a treadmill, starting four weeks before the lesion, followed by three days of rest and a week post-lesion. The training involved 5 days per week, with varying speeds and durations: Days 1–4: 7 min at 11 cm/s; Days 5–7: 7 min at 15 cm/s; Days 8–10: 7 min at 15 cm/s; Days 11–13: 10 min at 15 cm/s; Days 14–16: 10 min at 25 cm/s; Days 17–25: 15 min at 25 cm/s from 10:00 to 16:00 h.

### 2.5. Tunnel Walk Recordings

Kinematic recordings were taken on day zero and the third or seventh day post-injury. Videos from day 0 were considered as control groups (pre-injury), and those from days 3 and 7 were injured groups. Two synchronized cameras recorded left and right hindlimbs at 240 fps, with a 1280 × 720 resolution. Post-processing removed spherical distortion using a homographic matrix from four image points, as before. Steps were selected using the manual definition of instants corresponding to the beginning and end of the step. Displacement curves for the ankle and metatarsus points were manually annotated using custom software. Each step was analyzed separately (Appendix A).

### 2.6. Dissimilarity Factor and Vertical Displacement Analysis

Changes in the dissimilarity factor (DF) among groups were determined by comparing displacement curves and calculating differences using the Euclidean distance between normalized curve points on horizontal (X) and vertical (Y) axes, as described previously.
$${DF}_{<a,b>}=\frac{1}{200}\sqrt{\sum_{i=1}^{100}{\left(x_{a}(i)-x_{b}(i)\right)}^{2}+\sum_{i=1}^{100}{\left(y_{a}(i)-y_{b}(i)\right)}^{2}}$$
where *DF*_<*a*,*b*>_ is the squared error between every point of the normalized curves, defined as difference factor (*DF*); “*x_a_*(*i*) − *x_b_*(*i*)” is the difference between the coordinates in *x*, and “*y_a_*(*i*) − *y_b_*(*i*)” in *y* of every point in the graph, when comparing two steps (*a* and *b*); and “*i*” is the percent in the step cycle [7,21]. We averaged the vector Y components at each end of the normalized displacement curves for each group at 3 and 7 days post-injury (Appendix A). We compared this pattern comparison analysis by using a locally designed MATLAB script (https://ww2.mathworks.cn/help/install/ug/get-new-matlab-release_zh_CN.html).

### 2.7. Statistical Analysis

Firstly, the Kolmogorov–Smirnov test was used to determine data normality. All results are expressed as means ± SEM. For DF analysis, we analyzed the differences between the total value of X and Y graphically (horizontal displacement and vertical displacement in all groups) using a Kruskal–Wallis test with Dunn post hoc. For the vertical displacement analysis, using the T-test, we compared the control curve point by point versus the curve in the experimental group. We divided each curve into one hundred points. The curve represents the mean of the total steps. Values of *p* ≤ 0.05 (*), *p* ≤ 0.01 (**), *p* ≤ 0.001 (***), and *p* ≤ 0.0001 (****) were considered statistically significant. We conducted the statistical analysis using the Prism 9.0 software (GraphPad, La Jolla, CA, USA).

## 3. Results

### 3.1. Dissimilarity Factor (DF) in the Left and Right Metatarsus on the Third and Seventh Days Post-Injury with Resveratrol and Exercise

On the third day post-injury (dpi), resveratrol and exercise with resveratrol treatment returned the DF to the control value in the left and right metatarsus joints (Figure 1A,B). The third-day injury rats showed a DF in the left metatarsus compared to the control (Figure 1A) but not in the right (Figure 1B). The exercise group showed a significant DF in the left metatarsus in the 3 and 7 dpi groups (Figure 1A,C). On the seventh day, in the resveratrol group, the decrease was less marked than on the third day, but resveratrol plus exercise returned the DF to the control level (Figure 1C). The DF was higher in the exercise group in both the left and right metatarsus compared to the control group in the 3 and 7 dpi groups (Figure 1A,C,D).

### 3.2. DF in the Ankle Was Treated with Resveratrol and Exercise on the Third and Seventh Days Post-Injury

The injury group on the third and seventh days showed an increase in DF compared to the control group. This was more evident in the 3 dpi group (Figure 2A,C). In the exercise group, the left and right ankle DF differed in the 3 and 7 dpi groups versus the control group, mainly in the left ankle (Figure 2A,C,D). The resveratrol and exercise group versus the resveratrol treatment group did not show a change in DF in the 3 and 7 dpi groups, but they returned ankle joint movements to the control level (Figure 2A–D). The resveratrol with the exercise group showed a significant difference compared to the exercise group in both ankles on the third and seventh days (Figure 2A–D).

### 3.3. Exercise Modifies Vertical Displacement (VD) in the Metatarsus and Ankle Post-Injury

We analyzed the displacement of vertical and horizontal steps in the left and right joints. However, we did not find significant differences in horizontal displacement in the metatarsus, ankle, and knee (Appendix A). On the third day post-injury, VD differed by 14% of the step cycle in the left metatarsus compared to control and only by 1% after exercise. The Ex 3 dpi group differed by 12% compared to the injury group (Figure 3A). In the right metatarsus, VD was 32% in the control group and 0% in the Ex 3 dpi group on the third day. In the Ex 3 dpi group, VD was 23% compared to the injury group (Figure 3B). On the seventh day post-injury, VD changed by 35% in the left metatarsus for the control group and 40% for the Ex 7 dpi group (Figure 3C). The VD between the control and Ex 7 dpi group was 9%. In the right metatarsus, post-injury changes were 32% versus the control group, but the Ex 3 dpi group showed a difference of 23% (Figure 3B). On the seventh day post-injury, VD changed by 9% in the control group and 32% after exercise (Ex 7 dpi group); the difference was 4% compared to the control group (Figure 3D).

In the injury 3 dpi group, the VD of the left ankle was 81% compared to the control group. VD of the control group versus the Ex 3 dpi group was 30%. The VD of the 3 dpi group was 83% compared to the Ex 3 dpi group (Figure 4A). In the 3 dpi group, the VD in the right ankle changed by 83% during the step cycle for the control group. The VD of the right ankle of the control group versus the Ex 3 dpi group was 11%. The VD in the 3 dpi group was 63% compared to the Ex 3 dpi group. (Figure 4B). On the seventh day post-injury, the VD in the left ankle was 63% compared to the control group. For the VD in the control group, the difference was 30% versus the Ex 7 dpi group. The VD in the 3 dpi group was 82% compared to the Ex 7 dpi group (Figure 4C). The VD in the right ankle differed 62% for the 7 dpi post-injury group compared to the control group. The change in VD in the control group was 25% compared to the Ex 7 dpi group. The VD differed by 67% in the 7 dpi group compared to the Ex 7 dpi injury group (Figure 4D).

### 3.4. VD in Metatarsus and Ankle Joints after Resveratrol Treatment in the Penetrating Injury Model

On the 3 dpi in the left metatarsus, the VD changed by 14% compared to the control group. The resveratrol treatment group on the 3 dpi differed by 26% compared to the control group. The VD in the metatarsus with injury differed by 6% compared to the injury and resveratrol treatment (Figure 5A). The VD between the control versus 3 dpi group was 32% in the right metatarsus. Comparing the control and 3 dpi group with resveratrol treatment, the VD changed by 21%. The resveratrol treatment on the 3 dpi modified the VD by 9% versus 3 dpi (Figure 5B). In the 7 dpi group, the VD changed by 35% compared to the control group. The resveratrol treatment on the 7 dpi differed by 21% versus the control group. The VD of the 7 dpi group versus the 7 dpi with resveratrol treatment group changed by 13% (Figure 5C). In the right metatarsus, the change was 9% on the seventh-day post-lesion compared to the control group. The VD between the control and 7 dpi with resveratrol treatment group did not change. In the 7 dpi with resveratrol treatment group, the VD decreased to 5%, compared to the 7 dpi group (Figure 5D).

The VD of the left ankle in the control group compared to the 3 dpi changed by 81%, respectively. In the control group, the VD was 11% versus 3 dpi with resveratrol treatment. Compared to the 3 dpi group versus the 3 dpi with resveratrol treatment group, it changed by 35% (Figure 6A). The VD in the right ankle on the 3 dpi changed by 83% concerning the control group. In the 3 dpi with resveratrol treatment group, only 46% changed compared to the control group. In the resveratrol-treated rats after 3 dpi, the VD during the step cycle only differed by 49% compared to the 3 dpi group (Figure 6B). The VD of the left ankle in the control group changed by 63% compared to the 7 dpi post-injury group, respectively. In the control group, the VD changed from 63% to 18% compared to the 7 dpi with resveratrol treatment group. Compared to the 7 dpi group versus the 7 dpi with resveratrol treatment group, it changed by 64% (Figure 6C). The VD in the right ankle on the 7 dpi changed by 62% concerning the control group. In the 7 dpi with resveratrol treatment group, it changed by 12% compared to the control group. In the resveratrol-treated rats 7 dpi, the VD during the step cycle only differed by 52% compared to the 7 dpi group (Figure 6D).

### 3.5. VD in Metatarsus and Ankle Joints after Exercise and Resveratrol Treatment in Male Rats

The VD in the left metatarsus changed by 14% of the step cycle in the control group compared to the 3 dpi group. In the 3 dpi exercise and resveratrol treatment group, the VD differed by 14% versus the control group. Three days post-injury, the VD differed by 41% when comparing the exercise with the resveratrol treatment group (Figure 7A). In the right metatarsus, the 3 dpi group change was 32% compared to the control group. In the 3 dpi with exercise and resveratrol treatment group, the VD did not change compared to the control group. Comparing the VD of the 3 dpi group versus the 3 dpi with exercise and resveratrol-treated rats, the difference was 43% (Figure 7B).

In the 7 dpi group, the VD changed by 35% compared to the control group. For the 7 dpi exercise with resveratrol, the treatment group differed by 10% from the control group. Comparing the VD of exercised rats with resveratrol treatment versus the injury group, the difference was 37% (Figure 7C). In the right metatarsus, for the 7 dpi group, the VD changed by 9% compared to the control group. The 7 dpi after exercise with resveratrol treatment group saw a change of 6% compared to the control group. Comparing exercised and resveratrol-treated rats versus injury rats, the difference was 20% (Figure 7D).

The VD in the left ankle joint of the control group compared to the 3 dpi group changed by 81%, respectively. The VD did not change for rats 3 dpi with exercise and resveratrol compared to the control group. The 3 dpi and exercise with resveratrol treatment group differed by 72% compared to the 3 dpi group (Figure 8A). The VD in the right ankle on the 3 dpi changed by 83% versus the control group. The 3 dpi exercise and resveratrol group changed by 30% compared to the control group. The 3 dpi exercise and resveratrol group changed by 79% compared to the 3 dpi group (Figure 8B). On the left ankle, the 7 dpi group was modified by 63% compared to the control group. For 7 dpi exercise, the resveratrol group changed by 26% compared to the control group. For the 7 dpi exercise, the resveratrol group changed by 54% compared to the 7 dpi group (Figure 8C). The VD in the right ankle changed by 62% in the 7 dpi group versus the control group. The exercise and resveratrol treatment group changed by 9% compared to the control group. Comparing rats 7 dpi versus exercise and resveratrol-treated rats, the VD changed by 56%, respectively (Figure 8D).

## 4. Discussion

This study describes changes in the metatarsus and ankle in a rat model of penetrating hippocampus injury and the effects of resveratrol treatment and/or exercise. We report that resveratrol and/or exercise improve metatarsus and ankle kinematics. The ankle and foot play a crucial role in body support, and abnormal movements can impact leg and trunk balance, altering gait. The metatarsal–ankle joint complex’s range of motion is critical for locomotion kinematics. Penetrating brain injuries cause gait impairment, making walking restoration a primary goal [22,23].

The hippocampus and striatum are linked to locomotion, with some hippocampal neurons reflecting the animal’s speed [24]. Both movement execution and motor imagery share a network, including the striatum [25]. Our lab’s previous studies showed that a penetrating hippocampus injury alters vertical displacement in rat and mouse hind limbs. This suggests the hippocampus’s role in locomotion control, similar to human hippocampus focal stroke [26]. Thus, the hippocampus could benefit significantly from neuroprotection [2].

This study evaluated exercise and resveratrol individually to improve vertical displacement during the step cycle on the third and seventh days post-injury. Improvement patterns appeared over time on both sides. Exercise’s effect on locomotion post-TBI or stroke has been reported, even in a single session [27]. Resveratrol showed a similar improvement, except in the left metatarsus on the third day post-injury. The right-side injury delayed left metatarsus recovery. This was evident on the seventh day. This warrants further study. Combined exercise and resveratrol also improved vertical displacement in the metatarsal and ankle joints post-injury, suggesting different mechanisms. Exercise may promote neurogenesis, while resveratrol may have anti-inflammatory effects [28,29,30].

Spontaneous recovery is attributed to neuronal plasticity observed in murine injury models, also reported in other murine studies [31,32,33]. This spontaneous recovery likely contributed to movement improvements in this study. While exercise alone did not significantly improve the DF compared to the penetrating injury model, resveratrol showed a more significant effect in restoring the DF post-injury [34]. Resveratrol enhances endurance in high-capacity running rats and, combined with exercise, improves fatigue and exercise intolerance in mice with heart failure [35].

This study demonstrates the variable effects of resveratrol and exercise, with metatarsus movements showing different patterns. Combined resveratrol and exercise post-injury on the third and seventh days appear to have other effects, possibly due to horizontal or lateral displacement effects, warranting further study. Resveratrol significantly improves functional recovery post-spinal cord injury by promoting axonal regeneration and suppressing apoptosis [36]. However, changes in different joints and times, or left and right metatarsus, were not seen. Spinal cord central pattern generators (CPGs) function individually for each articulation and leg [37]. Altered muscle activation patterns may result from changes in spinal CPG function or altered higher structure information converging in spinal motor circuits. These CPGs can function autonomously post-denervation, highlighting the hippocampus’s role in CPG function [38,39,40,41].

Resveratrol is a potent neuroprotective compound with mechanisms including antioxidant action [42], reducing pro-inflammatory cytokines [43], preventing apoptosis [44], and decreasing microglial activation [45]. The hippocampus, heavily affected by injury, ischemia, or neurodegeneration [46,47], could benefit significantly from neuroprotection. Exercise is also effective for recovering lower extremity kinematic alterations post-penetrating hippocampus injury. Thus, resveratrol and exercise could be considered for treating hippocampal damage in various pathologies with motor implications.

## 5. Conclusions

This study shows that exercise and resveratrol contribute to improving locomotion kinematics following a penetrating hippocampal injury. Administration of exercise and resveratrol, individually or in combination, facilitates the recovery of vertical displacement in the metatarsal and ankle joints. The combination of both treatments exhibits a more pronounced synergistic effect, suggesting that exercise and resveratrol are beneficial and that their combined use may offer more comprehensive motor recovery. These findings underscore the potential of combined therapeutic strategies for the rehabilitation of brain injuries, highlighting the importance of exploring multifaceted approaches to enhance motor functionality in brain injury models.

## Figures and Tables

**Figure 1 brainsci-14-00980-f001:**
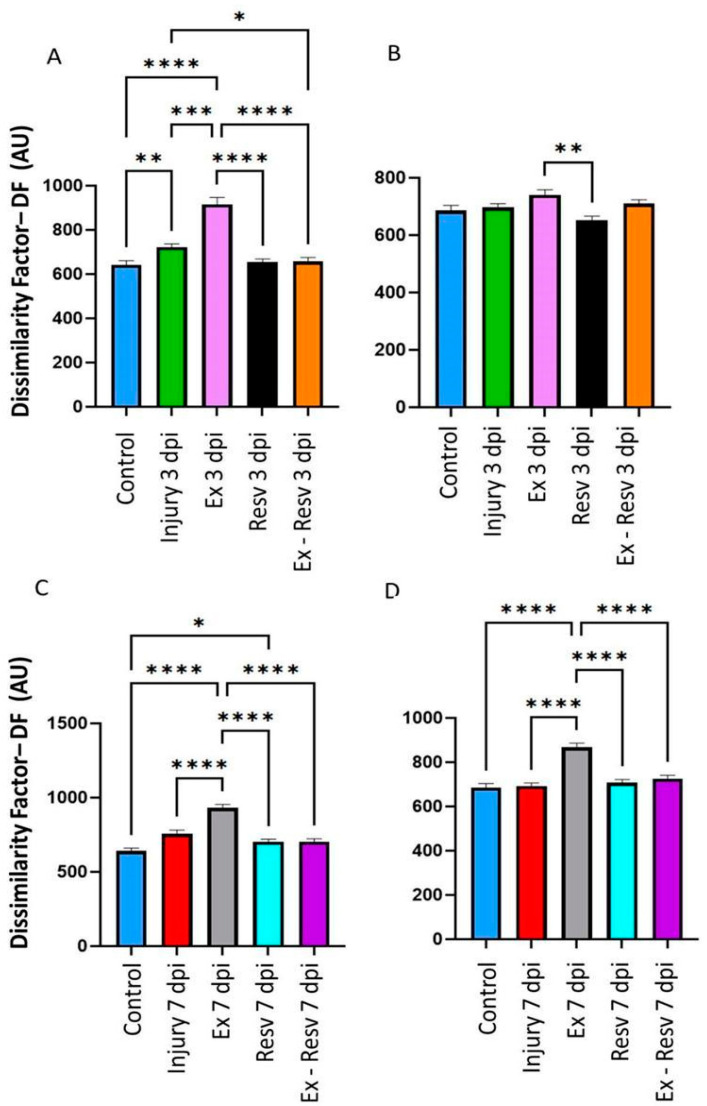
(**A,B**) Left and right metatarsus. Bar graphs illustrate the dissimilarity factor (DF) among the control and experimental groups in metatarsus. Control is the blue bar; injury at three days (injury 3 dpi) is illustrated as green. (**C**,**D**). Left and right metatarsus at seven days (injury 7 dpi) post-injury, red bars. Exercise (Ex) at 3 and 7 days post-lesion is illustrated in pink and gray bars, respectively; resveratrol treatment (Resv) at 3 and 7 days post-lesion is in black and blue bars, respectively; and exercise with resveratrol treatment (Ex-Resv) at 3 and 7 days post-lesion is in orange and cyan bars, respectively. The values are expressed as median ± SE. The asterisks illustrate statistical differences between groups utilizing the Kruskal–Wallis test. *p* ≤ 0.05 (*), *p* ≤ 0.01 (**), *p* ≤ 0.001 (***), *p* ≤ 0.0001 (****).

**Figure 2 brainsci-14-00980-f002:**
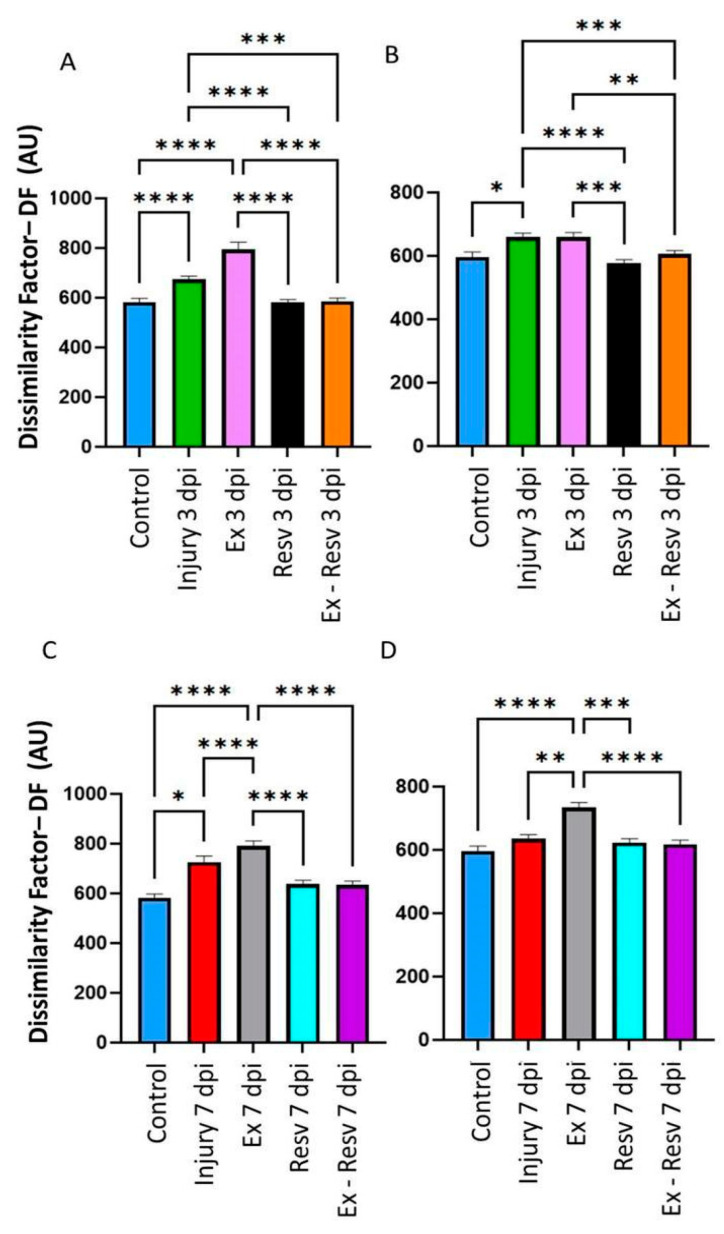
(**A**,**B**) Left and right ankle. Bar graphs illustrate the dissimilarity factor (DF) among the ankle’s control and experimental groups. Control, blue bar; injury at 3 days post-injury in green. (**C**,**D**), Blue bars illustrate control left and right ankle DF and red bars the DF at seven days post injury. Exercise (Ex) at 3 and 7 days post-injury, pink and gray bars, respectively; resveratrol treatment (Resv) at 3 and 7 days post-lesion, black and blue bars, respectively; and exercise with resveratrol treatment (Ex-Resv) at 3 and 7 days post-injury, orange and cyan bars, respectively. The values are expressed as median ± SE. The asterisks illustrate statistical differences between groups utilizing the Kruskal–Wallis test. *p* ≤ 0.05 (*), *p* ≤ 0.01 (**), *p* ≤ 0.001 (***), *p* ≤ 0.0001 (****).

**Figure 3 brainsci-14-00980-f003:**
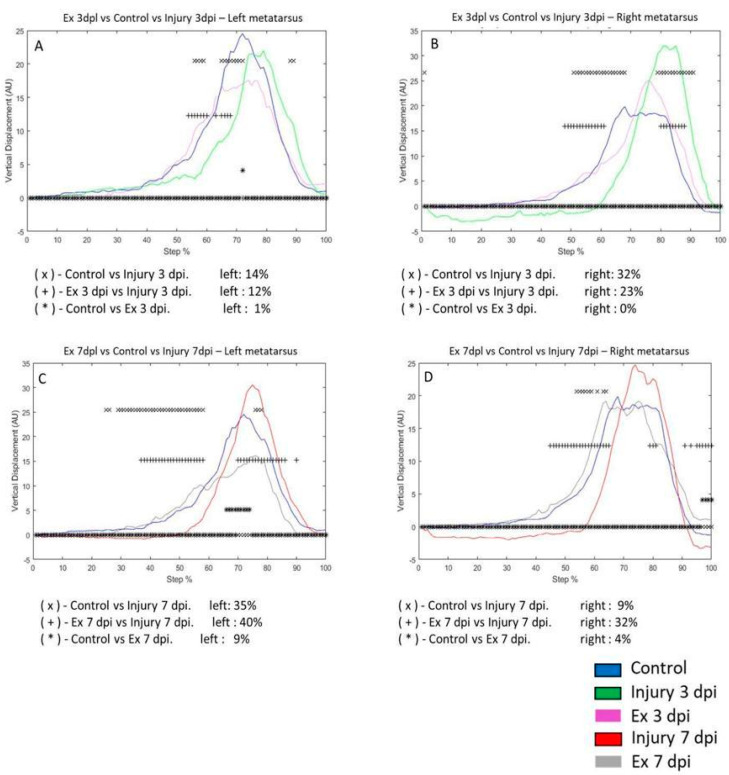
(**A**,**B**) The graphs illustrate the vertical displacement (VD) in the left and right metatarsus of the control (blue line), injury after three days (3 dpi) (green line), and exercise post-injury groups (Ex) (cyan line), respectively. (**C**,**D**) correspond to VD in the left and right metatarsus of control (blue line) injury after seven days (7 dpi) (red line) and exercise post-injury groups (Ex) (gray line), respectively. The asterisks illustrate the bins with a statistical difference (* *p* ≤ 0.05). The percent of change is expressed below the graphs.

**Figure 4 brainsci-14-00980-f004:**
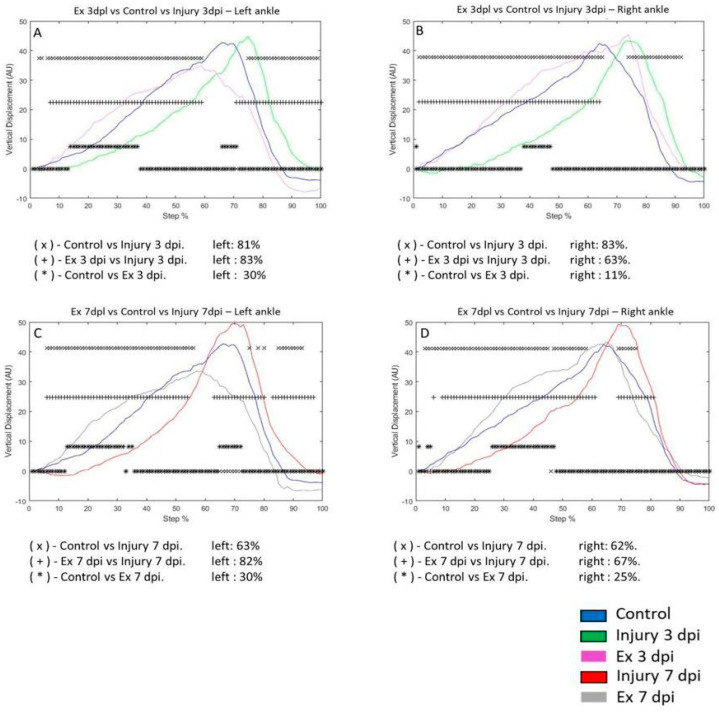
(**A**,**B**) The graphs illustrate the VD in the left and right ankle of the control (blue line), injury after three days (3 dpi) (green line), and exercise post-injury groups (Ex) (cyan line), respectively. (**C**,**D**) correspond to the VD in the left and right ankle of control (blue line) injury after seven days (7 dpi) (red line) and exercise post-injury groups (Ex) (gray line), respectively. The asterisks illustrate the bins with a statistical difference (* *p* ≤ 0.05). The percent of change is expressed below the graphs.

**Figure 5 brainsci-14-00980-f005:**
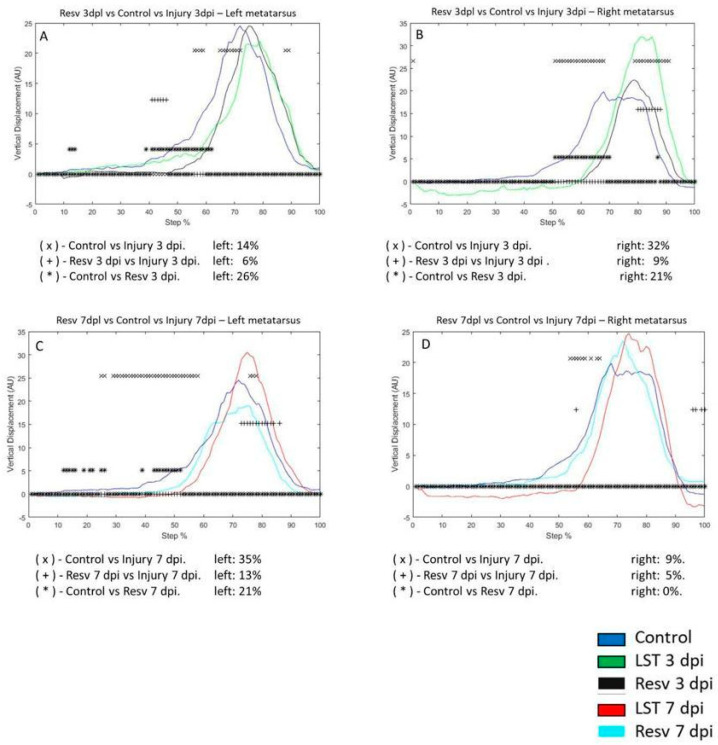
(**A**,**B**) The graphs illustrate the VD in the left and right metatarsus of the control (blue line), injury after three days (3 dpi) (green line), and resveratrol post-injury groups (Res) (cyan line), respectively. (**C**,**D**) correspond to the VD in the left and right metatarsus of control (blue line), injury after seven days (7 dpi) (red line), and resveratrol post-injury groups (Resv) (gray line), respectively. The asterisks illustrate the bins with a statistical difference (* *p* ≤ 0.05). The percent of change is expressed below the graphs.

**Figure 6 brainsci-14-00980-f006:**
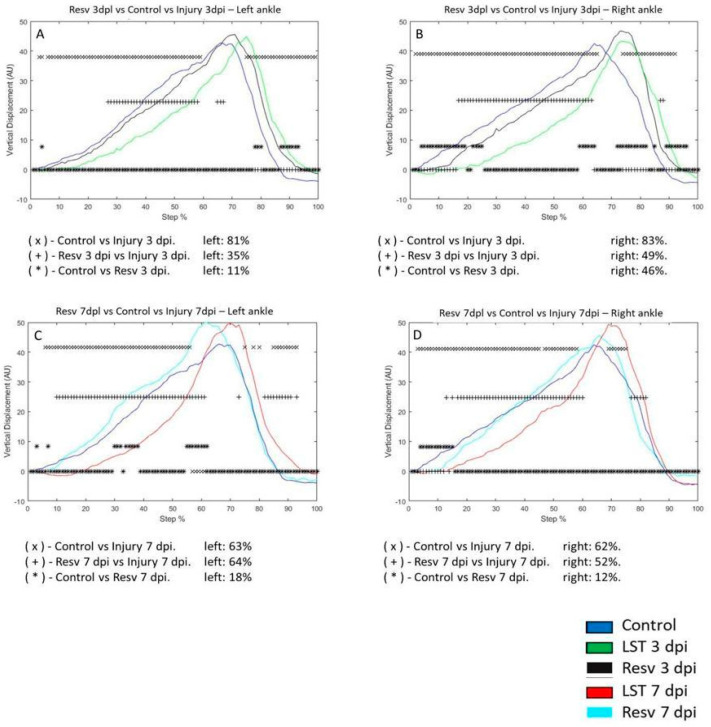
(**A**,**B**) The graphs illustrate the vertical displacement (VD) in the left and right ankle of the control (blue line), injury after three days (3 dpi) (green line), and resveratrol post-injury groups (cyan line), respectively. (**C**,**D**) correspond to the VD in the left and right ankle of the control (blue line), injury after seven days (7 dpi) (red line), and resveratrol post-lesion groups (Resv) (gray line), respectively. The asterisks illustrate the bins with a statistical difference (* *p* ≤ 0.05). The percent of change is expressed below the graphs.

**Figure 7 brainsci-14-00980-f007:**
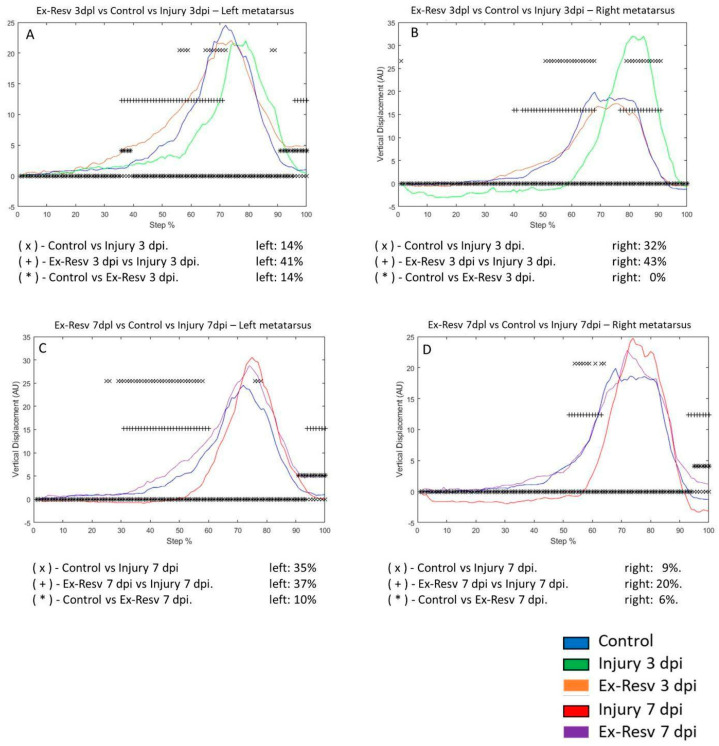
(**A**,**B**) The graphs illustrate the vertical displacement (VD) in the left and right metatarsus of the control (blue line), injury after three days (3 dpi) (green line), and exercise plus resveratrol post-injury groups (Ex-Resv) (cyan line), respectively. (**C**,**D**) correspond to the VD in the left and right metatarsus of the control (blue line), injury after seven days (7 dpi) (red line), and exercise plus resveratrol post-injury groups (Ex-Resv) (gray line), respectively. The asterisks illustrate the bins with a statistical difference (* *p* ≤ 0.05). The percent of change is expressed below the graphs.

**Figure 8 brainsci-14-00980-f008:**
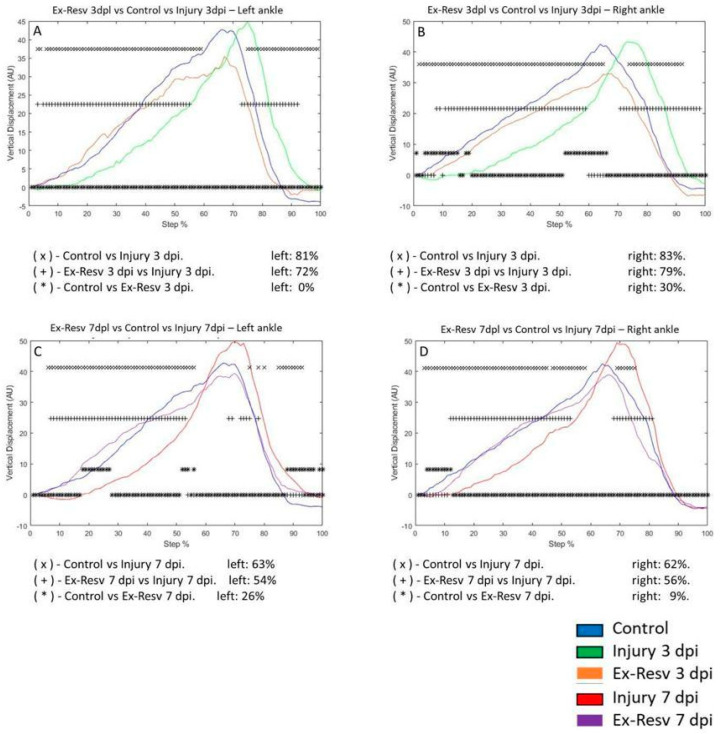
(**A**,**B**) The graphs illustrate the vertical displacement (VD) in the left and right ankle of the control (blue line), injury after three days (3 dpi) (green line), and exercise plus resveratrol post-injury groups (Ex-Resv) (cyan line), respectively. (**C**,**D**) correspond to the VD in the left and right ankle of control (blue line), injury after seven days (7 dpi) (red line), and exercise plus resveratrolpost-injury groups (Ex-Resv) (gray line), respectively. The asterisks illustrate the bins with a statistical difference (* *p* ≤ 0.05). The percent of change is expressed below the graphs.

## Data Availability

The raw data supporting the conclusions of this article will be made available by the authors by request without reservations.

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
