# Peer review of "Resveratrol and Exercise Produce Recovered Ankle and Metatarsus Joint Movements after Penetrating Lesion in Hippocampus in Male Rats"

_brainsci, 2024, doi:10.3390/brainsci14100980_

Round 1

Reviewer 1 Report

Comments and Suggestions for Authors

- Many references are too old and just 2 references belong to 2023. Use the new version instead of the old one.

- The number of animals per group is not clear.

- Did the authors prepare the animals for exercise before the main experiment? More details should be provided in the methods section.

- What is the basis for choosing the dose of Resveratrol?

- What are the criteria for allocating the animals in different groups?

- The study lacks a sham surgery control group.

- The basis for choosing statistical tests isn't always clearly stated.

- More direct evidence linking hippocampal function to lower limb kinematics would strengthen the study conclusions rather than connecting the observations to Alzheimer's disease.

- The exercise time is too short to induce physiological effects.

- It is not clear why these time points were chosen for applying the different treatments.

- There is no Resveratrol control group to exclude the effect of ethanol as a solvent.

- More details about how animal suffering was minimized should be provided.

- How does the present model accurately represent the clinical brain injuries in human?

Comments on the Quality of English Language

Minor editing of English language required.

Author Response

Thank you very much for taking the time to review this manuscript. Please find the detailed responses below and the corresponding revisions

  1. Many references are too old, and just 2 references belong to 2023. Use the new version instead of the old one.

We agree with this comment. References were updated

  1. The number of animals per group is not clear.

Thank you for pointing this out. The number of animals were 25, divided into five by group.

  1. Did the authors prepare the animals for exercise before the main experiment? More details should be provided in the methods section.

We mentioned it in section 2.4 of the exercise training. 

  1. What is the basis for choosing the dose of Resveratrol?

We are based on the author Sönmez U et al., 2007 (doi: 10.1016/j.neulet.2007.04.070)

  1. What are the criteria for allocating the animals to different groups?

We use the variables resveratrol dose, exercise,  their combination, and the lesson to compare with control status, evaluating their similarity factor as vertical displacement.   

  1. The study lacks a sham surgery control group.

We considered not using the sham group because Andreu M. et al., 2023, did not describe any changes in the sham group. This allowed us to reduce the number of animals employed in this study by following ethical rules. 

Andreu M, Sanchez LMQ, Spurlock MS, Hu Z, Mahavadi A, Powell HR, Lujan MM, Nodal S, Cera M, Ciocca I, Bullock R, Gajavelli S. Injury-Transplantation Interval-Dependent Amelioration of Axonal Degeneration and Motor Deficit in Rats with Penetrating Traumatic Brain Injury. Neurotrauma Rep. 2023 Apr 10;4(1):225-235. doi: 10.1089/neur.2022.0087. PMID: 37095855; PMCID: PMC10122235.

  1. The basis for choosing statistical tests isn't always clearly stated. 

We clarify the section 2.7.

  1. More direct evidence linking hippocampal function to lower limb kinematics would strengthen the study conclusions rather than connecting the observations to Alzheimer's disease.

Previously, we demonstrated the importance of hippocampus lesions with motor impairments, and this point refers to dementia diseases and stroke. You check the references 6 and 7.

  1. The exercise time is too short to induce physiological effects.

It is clear that on the seventh day with exercise, the vertical displacement is more similar in the intact group compared to the lesson group. Nevertheless, longer training with exercise could better affect the outcome using the dissimilarity factor or the vertical displacement. 

  1. It is not clear why these time points were chosen for applying the different treatments.

In a previous paper of our group, we found changes on the seventh-day post-injury. We were interested in studying whether these changes are similar on the third day post-injury

Alpirez J, Leon-Moreno LC, Aguilar-García IG, Castañeda-Arellano R, Dueñas-Jiménez JM, Asencio-Piña CR, Dueñas-Jiménez SH. Walk Locomotion Kinematic Changes in a Model of Penetrating Hippocampal Injury in Male/Female Mice and Rats. Brain Sci. 2023 Nov 2;13(11):1545. doi: 10.3390/brainsci13111545. PMID: 38002505; PMCID: PMC10669690.

  1. There is no Resveratrol control group to exclude the effect of ethanol as a solvent.

The ethanol dose does not produce changes in an experimental lesson because it is a low dose

 reported by: 

Yu, P., Wang, L., Tang, F. et al. Resveratrol Pretreatment Decreases Ischemic Injury and Improves Neurological Function Via Sonic Hedgehog Signaling After Stroke in Rats. Mol Neurobiol 54, 212–226 (2017). https://doi.org/10.1007/s12035-015-9639-7

  1. More details about how animal suffering was minimized should be provided.

The animals were raised under standard laboratory animal conditions according to the Mexican Regulation for the Care and Maintenance of Animals (NOM-062-ZOO-1999, 2001). The animals were kept under optimal conditions throughout the process, before, during, and after the treatment.  The animals were anesthetized with isoflurane during the surgical procedure and maintained on a 37°C bed. Post-surgical treatment included the administration of antibiotics and analgesics.

  1. How does the present model accurately represent the clinical brain injuries in human?

Several researches describe the importance of high morbidity and mortality in TBI, for example, a blow to the head or bullet penetration. They have the highest rates of TBI-related hospitalization and death. We demonstrated that penetrating injury in the hippocampus alters locomotion, which could impact the patients who suffer an injury and dementia pathology in the future. 

Abdelmalik PA, Draghic N, Ling GSF. Management of moderate and severe traumatic brain injury. Transfusion. 2019 Apr;59(S2):1529-1538. doi: 10.1111/trf.15171. PMID: 30980755.

Leung KK, Carr FM, Russell MJ, Bremault-Phillips S, Triscott JAC. Traumatic brain injuries among veterans and the risk of incident dementia: A systematic review & meta-analysis. Age Ageing. 2022 Jan 6;51(1):afab194. doi: 10.1093/ageing/afab194. PMID: 34651165.

Reviewer 2 Report

Comments and Suggestions for Authors

Review on the manuscript titled “RESVERATROL AND EXERCISE PRODUCE A RECOVERY ANKLE AND METATARSUS JOINT MOVEMENTS AFTER A PENETRATING LESION IN THE HIPPOCAMPUS IN MALE RATS” by Aguilar-García et al., 2024.

                The authors apparently investigated the effect of BRAIN (hippocampus) injury (Traumatic Brain Injury, TBI) impact on previously reported motor alterations in locomotion kinematics, and leg joints (ankle and metatarsal) parts in particular. The authors induced hippocampus lesions and observed the ankle/metatarsal kinetics abnormalities. In the course of the study, the authors observed the impact on the third and seventh day. Two phenotypic parameters were assessed in the study: the dissimilarity factor (DF) and vertical displacement (VD).

                The manuscript consists of Introduction, where the authors briefly explain/support their problem statement within the published materials. Next goes Material and Methods, corresponding to their research path: Penetrating Injury Model -> Resveratrol Treatment-> Exercise Training-> Tunnel Walk Recordings-> Dissimilarity Factor and Vertical Displacement Analysis.

                Results also may be outlined by corresponding chapters: ‘Dissimilarity Factor (DF) in the left and right metatarsus on the third- and seventh-days post-injury with resveratrol and exercise’ -> ‘DF in the ankle was treated with resveratrol and exercise on the third- and seventh-days post-injury’ -> ‘Exercise modifies vertical displacement (VD) in the metatarsus and ankle post-injury.’ ->’ VD in metatarsus and ankle joints after resveratrol treatment in the penetrating injury model’ -> ‘VD in metatarsus and ankle joints after exercise and resveratrol treatment in male rats’

After the study of compensating means for the TBI consequences in leg joints, the authors state: “Both exercise and resveratrol partially recovered VD in the ankle and metatarsus joints on these days”.

                While the study may be of interest for treating the consequences of TBI, it manifests purely phenomenological study without background molecular mechanisms elucidation. I strongly believe that the manuscript suits much more for pharmacology/physiology journal. Due to this issue, the authors don’t confirm which part of the cascade chain hippocampus-…- peripheral nerves are specifically targeted by exercise/resveratrol. So, it should be considered as a physiological/phenomenological study, and I’m not sure if this is Brain Science journal’s basic avenue.

1)      “A total of 27 adult Wistar rats were used”… “The same animals used as the control group were trained with exercise or treated with resveratrol or both on the third- or seventh-day post-injury.” – I’m not sure I can understand it.

I take it they were sequentially used along the study, but I doubt if it is valid/acceptable. While the control measurements may be obtained at the first round, the next steps exclude the possibility of sequential applications – only parallel ones.

2)      2.4. Exercise Training: “Exercise training was conducted on a treadmill, starting four weeks before the lesion” – what’s the purpose of training before the lesion session? Please, elaborate.

3)      Please, provide graphical timescale of experiments for 2.2-2.5 chapters.

4)      Please, provide the reference for your formula of DF (chapter 2.6) if it’s already been (used

5)      There is no formula for VD calculation in 2.6. It was mentioned in chapter 2.6 title, but never appeared within it. Please, address it.

6)      Why did the authors choose hippocampus, but not dorsal striatum, which is involved far more directly in movement activities? Please, elaborate on your choice of brain region.

7)      I haven’t found Supplementary Figure 1 within the draft.

8)      Why did the authors analyze days 3/7 only? What’s the reason for that? Was it complicated to assess all days of experiment?

9)      “Spontaneous recovery is attributed to neuronal plasticity observed in murine injury models, also reported in other murine studies [30,31,32]. This spontaneous recovery likely contributed to movement improvements in this study” – That may mean the current study have strong outliers disrupting the consistency of the conclusions. They should be treated accordingly.

10)   Please provide all the data you collected (on DF, VD) in supplementary for reproduction of the results.

Comments on the Quality of English Language

  The English is acceptable

Author Response

Thank you very much for taking the time to review this manuscript. Please find the detailed responses below and the corresponding revisions

1)      “A total of 27 adult Wistar rats were used”… “The same animals used as the control group were trained with exercise or treated with resveratrol or both on the third- or seventh-day post-injury.” – I’m not sure I can understand it.

We made a mistake with the number of animals, but the total was 25, divided into five by group. One group was considered a control and only filmed on the third and seventh day. All groups were filmed before drug administration or training with exercise. Thus, the second group was injured and filmed on the third and seventh day and then sacrificed. The third group was injured, administered with resveratrol, filmed on the third and seventh day, and then sacrificed. The fourth group was injured and trained with exercise, filmed on the third and seventh day, and then sacrificed. The fifth group was injured and administered with resveratrol and trained with exercise as well as filmed on the third and seventh day and then sacrificed. Therefore, we have a control in a group of intact rats and also a control per group. 

2)      2.4. Exercise Training: “Exercise training was conducted on a treadmill, starting four weeks before the lesion” – what’s the purpose of training before the lesion session? Please, elaborate.

The training aims to habituate the animals to walking at certain speeds so that they can later walk freely at their own pace with different stride lengths in the tunnel.

3)      Please provide a graphical timescale of experiments for 2.2-2.5 chapters.

Thanks for your commentary; please see supplementary Figure 1.

4)      Please, provide the reference for your formula of DF (chapter 2.6) if it’s already been used

Thank you for pointing this out. Please check the references: 

 Aguilar-García IG, Jiménez-Estrada I, Castañeda-Arellano R, Alpirez J, Mendizabal-Ruiz G, Dueñas-Jiménez JM, Gutiérrez-Almeida CE, Osuna-Carrasco LP, Ramírez-Abundis V, Dueñas-Jiménez SH. Locomotion Outcome Improvement in Mice with Glioblastoma Multiforme after Treatment with Anastrozole. Brain Sci. 2023 Mar 15;13(3):496. doi: 10.3390/brainsci13030496. PMID: 36979306; PMCID: PMC10046174.

Alpirez J, Leon-Moreno LC, Aguilar-García IG, Castañeda-Arellano R, Dueñas-Jiménez JM, Asencio-Piña CR, Dueñas-Jiménez SH. Walk Locomotion Kinematic Changes in a Model of Penetrating Hippocampal Injury in Male/Female Mice and Rats. Brain Sci. 2023 Nov 2;13(11):1545. doi: 10.3390/brainsci13111545. PMID: 38002505; PMCID: PMC10669690.

León-Moreno LC, Castañeda-Arellano R, Aguilar-García IG, Desentis-Desentis MF, Torres-Anguiano E, Gutiérrez-Almeida CE, Najar-Acosta LJ, Mendizabal-Ruiz G, Ascencio-Piña CR, Dueñas-Jiménez JM, Rivas-Carrillo JD, Dueñas-Jiménez SH. Kinematic Changes in a Mouse Model of Penetrating Hippocampal Injury and Their Recovery After Intranasal Administration of Endometrial Mesenchymal Stem Cell-Derived Extracellular Vesicles. Front Cell Neurosci. 2020 Sep 10;14:579162. doi: 10.3389/fncel.2020.579162. PMID: 33192324; PMCID: PMC7533596.

5)      There is no formula for VD calculation in 2.6. It was mentioned in chapter 2.6 title, but never appeared within it. Please, address it. 

We added the formula to material and methods.

6)      Why did the authors choose hippocampus, but not dorsal striatum, which is involved far more directly in movement activities? Please, elaborate on your choice of brain region.

 We were interested in the hippocampus's participation in movement activities because, in previous studies of our laboratory, we demonstrated kinematic changes in locomotion in hippocampus-injured animals:

López Ruiz JR, Osuna Carrasco LP, López Valenzuela CL, Franco Rodríguez NE, de la Torre Valdovinos B, Jiménez Estrada I, Dueñas Jiménez JM, Dueñas Jiménez SH. The hippocampus participates in the control of locomotion speed. Neuroscience. 2015 Dec 17;311:207-15. doi: 10.1016/j.neuroscience.2015.10.034. Epub 2015 Oct 24. PMID: 26597762.

In addition, the importance of the hippocampus in locomotion patterns has been demonstrated:

Bender F, Gorbati M, Cadavieco MC, Denisova N, Gao X, Holman C, Korotkova T, Ponomarenko A. Theta oscillations regulate the speed of locomotion via a hippocampus to lateral septum pathway. Nat Commun. 2015 Oct 12;6:8521. doi: 10.1038/ncomms9521. PMID: 26455912; PMCID: PMC4633825.

7)      I haven’t found Supplementary Figure 1 within the draft.

Thank you for your comment. It was added newly.

8)      Why did the authors analyze days 3/7 only? What’s the reason for that? Was it complicated to assess all days of experiment?

We have resulted in previous experiments in seven days in the lessoned hippocampus in an acute process. In a prior paper of our group, we found changes on the seventh day post-injury. We were interested in studying whether these changes are similar on the third day post-injury

Alpirez J, Leon-Moreno LC, Aguilar-García IG, Castañeda-Arellano R, Dueñas-Jiménez JM, Asencio-Piña CR, Dueñas-Jiménez SH. Walk Locomotion Kinematic Changes in a Model of Penetrating Hippocampal Injury in Male/Female Mice and Rats. Brain Sci. 2023 Nov 2;13(11):1545. doi: 10.3390/brainsci13111545. PMID: 38002505; PMCID: PMC10669690.

9) “Spontaneous recovery is attributed to neuronal plasticity observed in murine injury models, also reported in other murine studies [30,31,32]. This spontaneous recovery likely contributed to movement improvements in this study” – That may mean the current study have strong outliers disrupting the consistency of the conclusions. They should be treated accordingly.

Spontaneous recovery has been previously studied, but not with the kinematic parameters or the precision of the analysis used in this study. These values, such as DF and Vertical displacement, can even be applied in patients with injuries affecting the ankle and metatarsus and improve the measurements of these joint movements in various pathologies with motor implications.

10)   Please provide all the data you collected (on DF, VD) in supplementary for reproduction of the results. 

See the supplementary figure 3. 

Round 2

Reviewer 1 Report

Comments and Suggestions for Authors

Most of the comments were answered.

Comments on the Quality of English Language

 Minor editing of English language required.

Author Response

Thank you very much for taking the time to review this manuscript.

Comment: Minor editing of English language required.

Response: The manuscript has already been reviewed by a native English speaker who made changes to it.

Comment: Does the introduction provide sufficient background and include all relevant references?

Response: We improved the introduction

Comment: Are the methods adequately described?

Response: The methods have been expanded further and are supported in the supplementary file.

Reviewer 2 Report

Comments and Suggestions for Authors

   The authors have addressed the majority of the comments, but there are some editing inconsistencies: a) Figures in supplementary should be names Suppl. Figure 1, 2, 3; b) they should be correspondingly referenced in the text. Currently there is only 'Supplementary Figure 1' reference in the text, but apparently it doesn't correspond to the context. Please, arrange supplementary figures/references accordingly. Since the supplementary figures are supplied with the text, maybe you'd prefer to reference them as 'Supplementary chapter 1..3' in the title strings.

Author Response

Thank you very much for taking the time to review this manuscript.

Comments: The authors have addressed the majority of the comments, but there are some editing inconsistencies: a) Figures in supplementary should be names Suppl. Figure 1, 2, 3; b) they should be correspondingly referenced in the text. Currently there is only 'Supplementary Figure 1' reference in the text, but apparently it doesn't correspond to the context. Please, arrange supplementary figures/references accordingly. Since the supplementary figures are supplied with the text, maybe you'd prefer to reference them as 'Supplementary chapter 1..3' in the title strings.

Response: Based on your comment, we made the suggested change and also added the supplementary chapter to the corresponding section.

Supplementary Chapter 1: See 2.5. Tunnel Walk Recordings

Supplementary Chapter 2: See 2.6. Dissimilarity Factor and Vertical Displacement Analysis

Supplementary Chapter 3: See 3.3. Exercise modifies vertical displacement (VD) in the metatarsus and ankle post-injury